# 'Just because I'm old it doesn't mean I have to be fat': a qualitative study exploring older adults' views and experiences of weight management

Sarah E Jackson,[1] Linn Holter,[1] Rebecca J Beeken[1,2]

[1]Department of Behavioural Science and Health, University College London, London, UK
[2]Leeds Institute of Health Sciences, University of Leeds, Leeds, UK

**Correspondence to**
Dr Sarah E Jackson;
s.e.jackson@ucl.ac.uk

## ABSTRACT

**Objectives** The aim of this study was to explore older adults' beliefs about the appropriateness of weight management, and how their experiences and expectations of weight management have changed as they have got older.

**Design** Qualitative semistructured interview study.

**Setting** UK.

**Participants** Older adults (≥65 years) in the UK who had recent (<5 years) experience of trying to manage their weight (n=15; 12 women; 73% white British).

**Results** Data were analysed using thematic analysis. Emergent themes highlighted that weight remained a concern for many older adults, although having a high body weight was seen to be more acceptable at older than younger ages. Excess weight was reported to have negative consequences for health and well-being which participants felt could be alleviated by losing weight. Participants were motivated to lose weight for appearance and health reasons, but mentioned finding it harder to lose weight as they had got older and generally felt they had received limited guidance on weight management from health professionals.

**Conclusions** The views of our participants highlight the need for further research into safe and effective methods of weight loss for older people and indicate that advice and support from health professionals would be welcomed.

## INTRODUCTION

The last three decades have seen a substantial rise in the number of older people affected by overweight and obesity, as a result of increases both in the total number of older adults[1 2] and the proportion who are carrying excess weight.[3 4] Repeat cross-sectional data from the National Health and Nutrition Examination Survey in the USA and the Health Survey for England indicate that although obesity is becoming increasingly prevalent across all age groups, there is a trend for greater rises among older (≥55 years) than among younger adults, with levels reaching 40% (vs 35%) in the USA[5] and 32% (vs 23%) in England in 2010.[4] Based on recent trends, and allowing for the ageing populations in each country, a striking

## Strengths and limitations of this study

► This study offers insight into older adults' experiences of and attitudes towards weight management.
► The sample size was small but comparable with other qualitative studies.
► Self-selection bias may explain the generally positive attitudes towards weight management.
► Our findings do not reflect the views of non-white ethnic groups in which attitudes towards weight and ageing may differ.
► These results provide some encouragement for health professionals unsure about discussing weight management with older patients; indicating advice is likely to be well received.

simulation study projected that there would be a further 65 million obese adults in the USA by the year 2030, of whom 24 million would be aged ≥60 years, and an additional 11 million in the UK, of whom 3.3 million would be aged ≥60 years.[6] However even without further rises in obesity prevalence, the number of older adults affected by obesity looks set to grow as the population continues to age.[7]

For the older population, carrying excess weight comes with additional health risks. A substantial number of the medical complications associated with overweight and obesity become increasingly prevalent with age.[8] Around 80% of older people have at least one chronic health problem, and 50% have two or more.[9] Obesity also exacerbates the age-related decline in physical function: among older men and women, high body weight and excess body fat mass are related to increased physical dysfunction and disability[10] and strongly predict decline in functional status and future disability.[11–13] If past trends in obesity prevalence continue, the annual costs of obesity-related diseases are projected to rise by 13%–16% in the USA by 2030, of which 4% will be attributable to population ageing alone, and by 24%–25% in

the UK, of which 10% will be attributable to ageing alone.[6] An increase of this nature in the UK is a real threat to the future affordability of the National Health Service (NHS).[14]

Given the considerable burden of obesity among older people, one might expect weight management for older people to be an urgent public health priority. However, there has been relatively little research into how best to promote and achieve healthy weight loss among older adults, and whether weight loss is even indicated for those at older ages is a controversial issue that has attracted considerable attention. Although studies have demonstrated that older people (aged ≥65 years) can achieve significant weight loss,[15–19] losses of lean body mass and bone mineral density frequently accompany reductions in fat mass.[15–20] In addition, there is some evidence that the morbidity and mortality risk associated with excess weight decreases with advancing age.[21–23] As a result, there is uncertainty as to whether the benefits of losing weight outweigh the risks, and medical professionals are often reluctant to advise that older patients affected by obesity should lose weight.[24 25] However, studies that report associations between weight loss and mortality among older adults with obesity often fail to take into account intentionality of weight loss.[26–28] Intentional weight loss does not appear to be associated with increased risk of mortality.[26 29 30] Furthermore, a review on weight loss in adults with obesity aged ≥65 years found evidence that in spite of losses of fat-free mass and bone density, lifestyle interventions lead to positive changes in body composition, physical function and metabolic and cardiovascular risk.[31] Another review of the effectiveness of weight loss interventions for older adults with obesity concluded that despite the controversy, weight loss appears advisable and the authors encouraged healthcare providers to target older adults with obesity who could benefit from participating in weight loss interventions.[32]

Perhaps unsurprisingly in light of the limited evidence base on weight loss in the older population, little is known about older adults' own attitudes towards weight management. Studies comparing body image concerns of older and younger women suggest that body weight, and appearance in general, remain important as people get older.[33–37] In fact, as people get older and put on weight, their actual body image becomes increasingly discordant with their ideal body image and body dissatisfaction may even increase.[38] In addition, health concerns may become more salient with age, motivating older people to take action to tackle their weight.[39] Survey data indicate that losing weight is something that many older adults want, and are trying, to do. In a population-representative survey of British adults, older participants (≥55 years) were equally as likely as their younger counterparts to report a desire to weigh less, although they were only about half as likely to currently be attempting to lose weight.[40] The reasons for these seemingly discrepant findings have yet to be established.

A clear understanding of older adults' attitudes towards weight management, including beliefs about the appropriateness of weight management at older ages, is needed. Moreover, in-depth insight into older people's experiences of trying to manage their weight and expectations relating to weight management attempts, and how these change with age, is required. Understanding these concepts may aid in the development of successful interventions to promote weight control in this population. The present study therefore aimed to explore older adults' attitudes towards and experiences and expectations of weight management. A qualitative design was chosen because we were not seeking to test a hypothesis, but rather to obtain a rich source of information to better understand the rationale behind attitudes towards weight loss, and experiences and expectations of weight management at older ages.[41]

## METHOD

### Participants and recruitment

Our inclusion criteria were older adults (age ≥65 years) living in the UK, who had recent (<5 years) experience of weight management. We excluded individuals who had any cognitive impairment that might prohibit their ability to recall their weight management experience or to give informed consent (eg, dementia). We chose to use interviews rather than focus groups as we were interested in learning about people's individual beliefs and experiences, rather than determining a group consensus. We did not want group discussions or concerns that others might view their beliefs to be 'incorrect' to influence individuals' accounts of their own unique experiences.

The study was advertised via posters and flyers displayed in and around University College London and at charities and local community centres that hold targeted activities for older adults. Interested and eligible participants were given an information sheet and the opportunity to ask questions. We offered the option of conducting the interview face-to-face (at the university) or over the telephone to encourage individuals to take part who might have otherwise been put off by a lack of flexibility around time (eg, because of work or personal commitments) or location (eg, because of distance or ability to travel), or difficulties in hearing accurately on the telephone. We aimed to recruit until it was felt that saturation had been reached (ie, sampling more data would not provide more information or insights related to our research questions). Based on previous qualitative studies in similar groups, we expected that around 15 participants would be required to achieve this,[42 43] although no target number of participants was set. All interviews were conducted with only the participant and interviewer present.

### Data collection

Sociodemographic questions covered age, sex and ethnic group. The preinterview questionnaire also asked participants to report their height and weight, which were used to calculate their body mass index (BMI; weight in kg divided by the square of height in metres). Participants who attended the interview in person were offered use of scales at the university to complete or verify their self-reported data, but this was not a requirement for participation.

Semistructured interviews were carried out by a female researcher (LH; BSc Psychology), an MSc Health

**Table 1** Topic guide for qualitative interviews

| Topics | Prompts |
| --- | --- |
| Introductions and background | |
| Introductions | Who we are and aim of study. Check length of interview (30–60 min) ok. |
| Experiences of weight management | |
| Weight history | Brief overview of weight across the life course. Always struggled with weight? |
| Recent attempts at weight loss | Methods. Success. |
| Any changes with age | Methods. How easy it is. Success/achieving goals. Reasons (or perceived reasons) for any changes. |
| Motivation to lose weight | |
| Reasons for losing weight | Appearance reasons. Health reasons. Doctor's advice. Family and friends. |
| Any changes with age | Amount of motivation. Reasons for losing weight. In what way? |
| Personal importance of weight | Impact on life/health/well-being. Has it changed with age? |
| Barriers to losing weight | |
| Current barriers | Lack of willpower. Lack of support. Financial issues. Health problems. Social life. |
| Any changes in barriers with age | |
| Appropriateness of weight management for older people | |
| Own views | Something you can/should be doing? |
| Perception of other people's views | Something you can/should be doing? Is it something family/friends encourage? Guidance from health professionals |

Psychology student who had completed qualitative training as part of her BSc and MSc. The interviewer had no previous relationship with the study participants. Participants were aware that the study was part of the interviewer's research degree. Interviews lasted approximately 1 hour, and were recorded and transcribed verbatim. A topic guide (table 1) was developed collaboratively by all authors to guide the interviews and comprised a series of open questions covering experiences of weight management and motivators and barriers for weight loss, with a focus on any changes that have occurred with ageing, as well as views on the appropriateness of weight management for older people. The interviewer was trained to have minimal verbal input and prompt only when appropriate.[44]

### Patient and public involvement

Patients were not involved in the development of the research questions or choice of study design. The topic guide was piloted with two lay individuals for acceptability and comprehension, and to provide an indication of the time required to participate.

### Analysis

Data were analysed using thematic analysis, a qualitative method for identifying, analysing and reporting themes.[45] This method was chosen with the aim of providing a rich description of the data, and to identify themes at an explicit level using a realist approach.[45] As data were acquired, SEJ read the transcripts for essential familiarisation and to generate initial codes. These were amended and refined through discussion between the researchers in an iterative process until a final coding list was agreed. SEJ coded all the transcripts according to the established coding structure in NVivo V.10 and a random selection of transcripts (n=2) were coded by a second researcher (RJB) to check for reliability. Inter-rater reliability for the coding was generally high (81%–100% agreement; mean 98%) with any minor differences resolved by discussion. SEJ and RJB collated codes into common themes, which were reviewed and refined, named and each given a written description. All themes were checked against the transcripts to ensure that they reflected the majority of participants. Individual experiences were also highlighted. The completed consolidated criteria for reporting qualitative research checklist is available in online supplementary material 1.[46]

### RESULTS
#### Participants

Fifteen interviews were conducted with 12 women and 3 men, aged between 67 and 84 years (table 2); 11 face-to-face at the university and 4 by telephone. After the target number of 15 interviews was completed, the authors discussed the themes emerging and agreed that saturation had been reached.[47] Participants were on average 71.1 years old, and the majority (73%) described their ethnicity as white British. While all reported having tried to manage their weight in the last 5 years, at the time of interview 33% had a BMI in the obese range ($\geq 30 \, kg/m^2$), a further 53% were overweight (BMI 25–29.9 $kg/m^2$) and the remaining 13% had a BMI at the upper end of the normal weight range (18.5–24.9 $kg/m^2$).

### Themes

Six themes emerged from the data: (1) mixed views on the importance of weight at older ages, (2) excess weight

**Table 2** Sociodemographic and health characteristics

| Sociodemographic details | Total sample (n=15) |
|---|---|
| **Sex, n (%)** | |
| Male | 3 (20.0) |
| Female | 12 (80.0) |
| **Age (years), mean±SD (range)** | 71.67±4.32 (67–84) |
| **Ethnicity, n (%)** | |
| White British | 11 (73.3) |
| White Irish | 1 (6.7) |
| White other | 3 (20.0) |
| **BMI, mean±SD (range)** | 30.15±5.43 (23.29–40.38) |
| Weight status, n (%) | |
| Normal weight | 2 (13.3) |
| Overweight | 8 (53.3) |
| Obese | 5 (33.3) |

BMI, body mass index.

is more acceptable at older than younger ages, (3) excess weight has a negative impact on health and well-being, (4) appearance and health are important motivators for weight loss, (5) losing weight gets harder as you get older and (6) limited guidance on weight management from health professionals. On the whole, there were no notable differences in responses by age, sex or weight status, so results are presented for the whole sample.

### Mixed views on the importance of weight at older ages

Participants spoke about the current importance of weight to them relative to when they were younger. Most participants felt weight control was just as important at older ages: 'Just because I'm old [it] doesn't mean that I have to be fat … It's very important to me, weight. It hasn't lost any importance at all' (106, female, 71 years, BMI 24.2) and 'Once you know you're overweight and you feel a certain way you never lose sight of that' (110, female, 71 years, BMI 40.4) and 'I don't think you ever get rid of that psychological fear of fat, or at least I don't know anyone who suddenly said it doesn't matter' (113, female, 71 years, BMI 35.7).

For some participants, the issue of weight control had become more relevant with age. A few people spoke about how they had gained weight gradually over the life course: 'When I look at my life, in terms of weight I was probably nine and a half stone until I was in my thirties and had my children. Then it went up to probably ten stone. And then in my fifties bordering on eleven stone. And then in my sixties eleven and a half stone' (102, female, 68 years, BMI 27.0) and 'As years progress, you put two pounds on there, a pound here, another stone, another stone. And I'm the biggest I've been now' (103, female, 69 years, BMI 26.1). Some reported having gained weight as they got older due to changes in lifestyle: 'I used to do a lot of exercise. As I got older, I stopped doing so much

so I got fatter' (105, male, 72 years, BMI 28.9). Others attributed age-related weight gain to specific health problems or medication use: 'For most of my life I've been a reasonable weight. … I had hip problems when I reached about 60 and had to have a hip replacement, and I put on a lot of weight before [the operation] because obviously I could not move as fast' (101, female, 67 years, BMI 28.2) and 'I've always been aware of what I eat but it doesn't seem to matter; I keep gaining and gaining and then having to really hold it back. And I think also now, that I am on medication—that that has a role to play in it as well' (110, female, 71 years, BMI 40.4).

However, several participants reported feeling less motivated to lose weight now that they were older: 'I think when you're young, the motivation is there. You want to get the latest fashion, and you want to look stunning. And you want to find a mate. When you're older you get your old mate and the motivation is gone' (102, female, 68 years, BMI 27.0). A couple questioned whether trying to lose weight at older ages was worth the effort: 'It can be done it's just you get quite lazy as you grow older. You think, 'Oh, is it worth it?'' (103, female, 69 years, BMI 26.1) and 'Sometimes you think, 'Oh well, what is the point?'' (115, female, 60 years, BMI 33.9) and 'My sister, she struggles with weight although she is quite slim…and on the phone this morning she said, 'Since my party I haven't been able to lose weight, I'm putting it on again'. And then she said, 'But at our age really, do we care at 80 and 71?' And when I think of it like that, I think, 'Well do I? Do I really care?' And then the next part of me says, 'Yes, you do'' (106, female, 71 years, BMI 24.2).

Participants generally felt that people shouldn't be concerned about age when it comes to losing weight: 'I don't think you should start thinking like that, because as soon as you start thinking 'I can't do this because I'm…' whatever, you start limiting yourself. And why should you? … So you take no notice when people sort of, when my children say 'You can't do that!' because I can!' (101, female, age 67, BMI 28.2). However, many mentioned taking a more laid back approach to weight loss: 'The older I got the, the more sensible I got I think. So I could do with being a stone lighter but, you know, I'm not desperate about it' (113, female, 71 years, BMI 35.7) and 'You have to recognise your limitations as you get older. I'm not doing too badly really' (111, male, 84 years, BMI 27.0). Participants also reported setting less ambitious weight loss goals: 'As far as I'm concerned, the best is that I have a moderate amount of weight loss that suits me until I get to a point where I think I look ok, and then I'll stop and I don't go much further' (101, female, age 67, BMI 28.2) and 'You've always got to look nice, but certainly when you're my age, over 70, I don't think you should try to be slim. I don't think it's healthy' (107, female, 78 years, BMI 28.0).

While most participants endorsed benefits of losing weight, one talked about the importance of maintaining body weight in order to have reserves in the event of illness: 'I firmly believe that there is nothing wrong with

older people being a bit fat-ish … to be honest I think do we not, as we get older, need to be a little more rounded?' (107, female, 78 years, BMI 28.0); but only to a certain point: 'I think if you're say 18 or 20 stone or something like that when you're 60, it's time to say get yourself a gastric band because it's dangerous' (107, female, 78 years, BMI 28.0).

### Excess weight is more acceptable at older than younger ages

Although participants generally considered weight to be an important issue, many reported feeling less social pressure to lose weight compared with when they were younger: 'There's no pressure at this point; I don't really feel I'm pressured to hurry up and get rid of another stone. I think I will eventually' (113, female, 71 years, BMI 35.7) and 'I think if you're older you [lose weight] for yourself, you don't do it to satisfy people around you, to be liked more. You just do it because you want to lose weight. … You do it because you feel you should do it, and for your own benefit, that's all' (103, female, 69 years, BMI 26.1) and 'You've got to do it yourself because it is something that you want; it's for you and you're not doing it for anybody else' (109, female, 72 years, BMI 26.4).

Several people talked about how their family and friends had tried to discourage them from worrying about their weight and raised concerns about weight loss: 'My daughter, who is wonderful in every and each way, says, 'Mum, just buy it, enjoy wearing it, you would look lovely—don't worry about your weight!" (102, female, 68 years, BMI 27.0), 'I have thinner friends who say, 'Just forget about it, you're alright', and I say, 'I can't!" (110, female, 71 years, BMI 40.4) and 'If you lose weight then they say, 'You don't look too good', 'He's looking very pale', and all of that' (105, male, 72 years, BMI 28.9).

Participants spoke about there being less stigma around overweight at older ages: 'I think a lot of people joke about it and say, 'Oh, it's middle age spread,' so they accept having more weight. I think there is less stigma than at a younger age' (110, female, 71 years, BMI 40.4) and '[There's] much less [stigma] because we become invisible. Nobody notices older people, or we all look alike or something. I mean I'm not exactly small and I've been walked into by people who just didn't see me. You become invisible, so nobody's going 'urgh'; you're just another old person with white hair and that's that' (113, female, 71 years, BMI 35.7).

### Excess weight has a negative impact on health and well-being

Participants talked about the physical health consequences of excess weight: 'The extra weight affects my sleep, my breathing is more affected, my joints are more affected' (110, female, 71 years, BMI 40.4) and 'I do get tired easily, and I think that's probably because I'm carrying more weight' (111, male, 84 years, BMI 27.0). However, more so, the focus was on the impact of weight on mental well-being. Several people talked about feeling depressed about their weight: 'It is pretty depressing looking at yourself in the mirror and thinking 'Oh God,

look at that!' … it's mental as well as physical' (103, female, 69 years, BMI 26.1) and 'I tried some things on this morning and I actually looked at myself and cried. I do get depressed with my weight; I don't like it, and I berate myself quite a bit as well. … I talk to myself going up and down the flat: 'Oh you fat cow, for god's sake have a bit of control" (106, female, 71 years, BMI 24.2). Some mentioned negative effects on self-esteem: 'I try hard not to be negative about my body, but if I'm honest I am. I don't feel as confident with it, it affects my self-esteem' (110, female, 71 years, BMI 40.4) and feelings of self-blame: 'I get annoyed with myself because I know it's my own fault that I'm putting on weight, that I'm not losing it' (105, male, 72 years, BMI 28.9).

Participants generally felt that losing weight would help to improve their health and well-being: 'I don't look huge… but oh, I'd love to be a size 14 again, and just be happy in my own skin' (102, female, 68 years, BMI 27.0). Those who had successfully lost weight reflected on how it had improved their physical functioning: 'I feel stronger with my weight down' (112, female, 70 years, BMI 23.3) and 'I've lost a lot and I do feel better about it. See I can get up now, I can bend down like I couldn't before, too – I couldn't even see my shoelaces. … You just feel so much better because you can do things' (109, female, 72 years, BMI 26.4). Psychological benefits were also mentioned: 'I think [losing weight] made me feel better about myself. It's confidence, it's a nice place to be. You think, 'Alright, I don't look too bad.' … [It] puts you in a nice frame of mind' (104, female, 74 years, BMI 28.0) and 'I feel better, lighter – more, how can I explain? Full of life' (103, female, age 69, BMI 26.1).

### Appearance is an important motivator for weight loss

Participants talked about appearance as one of the primary reasons for wanting to manage their weight. For many, appearance was a powerful motivator: 'Vanity stays with you all your life if you've always been a bit attractive. I like to keep it that way…obviously it's gone, it's going, but I don't want to look a mess for my sake, for my children's sake if they see me' (107, female, 78 years, BMI 28.0) and 'I'd like to look good. I'd like to look how I think I look, and I think I look 6 foot 2, broad shoulders, narrow waist, hips and all of that. When I look in the mirror I know I don't' (105, male, 72 years, BMI 28.9) and 'I know it sounds stupid, but [health] takes second place to appearance. It shouldn't do, but it does' (106, female, 71 years, BMI 24.2).

However, not everyone was concerned about losing weight for appearance reasons: 'I'm quite confident. I see other people wearing beautiful clothes and would love to get into them, but it doesn't worry me that I can't really. … It's sort of not a major issue for me' (102, female, age 68, BMI 27.0).

### Health is an important motivator for weight loss

The majority of participants cited health as a key driver for weight loss and spoke about health concerns becoming

increasingly important as they had got older: 'I suppose it started in my fifties. I noticed that I could not control [my weight] very easily. And then in the last 6–7 years when I've been diagnosed with a lung problem' (110, female, 71 years, BMI 40.4) and 'I think it's more dangerous [to be overweight] when you're older, because of stroke and heart attacks and things like that' (109, female, 72 years, BMI 26.4). For a few, appearance had taken a backseat to health as they had got older: 'I don't care so much [about appearance] now, it's more the health side of it. When you get older, you don't care a damn what people think of you; not like when you're younger' (114, female, 70 years, BMI 36.5) and 'Fat's no longer a fashion issue, but it's become more of a health issue' (113, female, 71 years, BMI 35.7).

Several participants spoke of specific health conditions that they felt could be improved by losing weight: 'I have breathing problems and so any extra weight makes breathing more difficult' (110, female, 71 years, BMI 40.4) and 'I've got a dodgy heart and I've got terrible arthritis. I've got a bad back, I've got bad knees and everything else. And you know for years the doctors have been saying to me: lose weight, it will be better … And before I lost three and a half stone I was borderline diabetic, so of course that was another factor that made me lose weight' (106, female, 71 years, BMI 24.2). One participant mentioned that weight loss could reduce the need to take medication to control weight-related health issues: 'My blood pressure is borderline high and I know that if I get rid of that extra stone it [will] probably come down into comfortable normal without any medication' (113, female, 71 years, BMI 35.7). Another spoke of fear of adverse consequences of weight-related health conditions: 'I gained a lot of weight up to two to three years ago. I was about 19 stone and I thought, 'This can't go on.' And because I'm diabetic, I thought, 'I've got to lose weight, I don't want to have my legs chopped off. I don't want to go blind. I've got to do something about this,' and I've lost about two and a half, three stone' (114, female, 70 years, BMI 36.5).

### Losing weight gets harder as you get older
Participants spoke unanimously about finding it more difficult to manage their weight as they got older: 'It's hard to lose weight when you get older, it's much harder. You can lose seven pounds in a week years ago just like that, but once you get to a certain age it's a lot harder' (114, female, 70 years, BMI 36.5) and 'When you're older it doesn't come off quite as quickly' (113, female, 71 years, BMI 35.7) and 'It appears to me, having looked at all my contemporaries, you do put on weight more easily [as you get older]. But what's more interesting, which I didn't expect to find—because no-one expects ageing to be interesting, but it actually is—is that you put on weight in different places' (101, female, age 67, BMI 28.2).

Multi-dimensional barriers to managing weight effectively at older ages were reported.

### Health-related barriers
The ageing process was reported to negatively influence ability to lose weight. A general 'slowing down' with age was commonly mentioned: 'I think it has to do with life-style, 'cause obviously you slow down—you know, it would be very abnormal if you didn't—and so I don't move as quick. If I clean a room in the house it used to take me half an hour, now it takes me an hour and a half' (104, female, 74 years, BMI 28.0) and 'See a young person could get up in the morning and run around or could do many more things. But when you're old, you go home, you get tired' (109, female, 72 years, BMI 26.4).

In addition to an impact on lifestyle, some participants speculated that this slowing down process might also have an effect on metabolism: 'It may be something to do with our makeup as we get older and metabolism slows down I would imagine; everything else slows down' (104, female, 74 years, BMI 28.0) and 'I don't know whether your metabolism just doesn't function as well as it did. So it's probably just a factor of ageing and organs that nothing works as efficiently as it did 50 years ago' (113, female, 71 years, BMI 35.7) and 'It's just a way of…how you digest your food I expect. I don't know' (103, female, age 69, BMI 26.1).

While health was a commonly reported motive for weight loss, specific problems with health and mobility were also mentioned as barriers to weight loss at older ages: 'For an older person it's very hard for them to lose weight because they can't get about like a young person anymore. And they do tend to put more weight on…you're not going to burn off the calories because you can't get about' (109, female, 72 years, BMI 26.4) and 'I'm not as mobile as I was. I started doing tai chi and pilates but it's not really exercise as such; I can't do that because of my hips. That's [got] a lot to do with my weight—I mean, I do use it as an excuse as well—but it is not just an excuse' (106, female, 71 years, BMI 24.2). Medication use was also thought to make it more difficult to lose weight: 'Huge amounts of older people are very isolated, and they're depressed and taking anti-depressants which are putting on the weight. It doesn't help to take off weight, depression' (110, female, 71 years, BMI 40.4).

### Emotional barriers
A commonly cited barrier to weight loss was a lack of willpower: 'I've got the motivation, I just haven't got the will. I think it's my willpower' (105, male, 72 years, BMI 28.9) and 'It is very difficult, staying motivated—the hardest thing' (110, female, 71 years, BMI 40.4). For some, an awareness of their own mortality and wanting to enjoy life made it hard to sustain their resolve to lose weight: 'I've lost that steely determination that I did have at one stage, you know: 'I'm going to do it'. … Now I think, 'Oh yes, I'm going to lose half a stone' and then we're invited to lunch and I think, 'Oh, stop it!', you know, friends of mine are dying so why should I give up [eating out]?' (102, female, age 68, BMI 27.0) and 'I know it sounds depressing—I'm enjoying my life, I'm very happy at the

moment—but there's no future in being old. Let's face it, you have to be realistic here, so you get to the stage where [you think], 'Well, if I can't enjoy going out for a drink at night, enjoying the food I like, what is there in life?" (107, female, 78 years, BMI 28.0).

A few participants mentioned comfort eating and using food to deal with loneliness: 'I don't know whether there is a little bit of comfort eating coming in maybe. I need something when my son and his grandchildren and his family are abroad; it just makes me a bit sad. And I'm not as active as I was before and that probably makes me a bit sad. So I think there is an element of comfort eating' (102, female, age 68, BMI 27.0) and 'Old age is very lonely, so you have lots of time. … Generally I'll only be out of my house three hours and that's a lot for old people…so you've got the rest of the 21 hours of the day left' (107, female, 78 years, BMI 28.0).

### Situational barriers

Other reported barriers to effective weight management predominantly related to retirement and the impact it had on free time, physical activity and disposable income. Some participants commented that retiring left them more time for food-focused social occasions: 'Now I'm retired and at home, we have a very active life. I don't mean active in exercise terms, but we've got lots of friends and our pleasure—I suppose one of the pleasures—is that you go out to nice places to eat and entertain, and everything is food-focused' (102, female, 68 years, BMI 27.0). Having more free time was also cited as an opportunity to be more active: 'Loads of people I'm older with are extremely active; even more so than people in their sort of middle years when they have children and don't have time to do things for themselves quite as much' (101, female, 67 years, BMI 28.2), but some commented that stopping work had made them more sedentary: 'Because I am retired now I have more time to sit, which I have never had really; I worked all my life' (104, female, 74 years, BMI 28.0) and 'I'm retired now so I'm not walking around all day. … I have to think about being active' (110, female, 71 years, BMI 40.4). Some believed a decline in physical activity was a central factor in age-related weight gain: 'I've had a weight problem all my life, but an awful lot of people who haven't suddenly find they've started gaining weight in their 50s, and that's essentially because of the inactivity. Men are really bad at this. Men go to work, then they retire, and then they sit. They probably don't eat much more but they do a whole lot less' (113, female, 71 years, BMI 35.7). Free bus passes provided to the over-60s were mentioned in the context of discouraging physical activity: 'The Freedom pass is another thing of concern, which I shouldn't have because it tends to make me hop on a bus for a couple of stops instead of walking' (105, male, 72 years, BMI 28.9). Retirement also had an effect on disposable income available to fund weight loss efforts: 'WeightWatchers is quite expensive; certainly when you're on a pension I find it's quite

dear. When you're at work you don't think about it' (104, female, 74 years, BMI 28.0).

While most explanations for weight loss becoming more difficult over time related to the ageing process, some participants attributed increasing difficulty in managing their weight to changes in the food environment: 'I think one of the big changes is [that now] you are bombarded with food. When I was growing up, you'd never think of going to get a takeaway. Where I live, on the high street, at every corner is something, it's all fast food' (110, female, 71 years, BMI 40.4) and 'I lived in a house where there wasn't actually a huge amount of food at some points when I was small, so not having enough to eat was relatively common' (101, female, 67 years, BMI 28.2). However, some commented that information on healthy living was more readily available than when they were younger: 'There was no information at all on what made you fat; nobody said anything about it (110, female, 71 years, BMI 40.4) and 'Everybody, absolutely everybody knows [what they should be eating]; I can't believe that anybody on the planet who doesn't know. The five a day message has certainly got around' (113, female, 71 years, BMI 35.7), and one participant mentioned that better availability of and access to healthy foods had made it easier to lose weight: 'I think it's better now, because I'm eating better things. And when I was trying to lose weight before there wasn't the variety of food available' (112, female, 70 years, BMI 23.3).

### Weight loss strategies less effective than at younger ages

In addition to these barriers, several participants commented that strategies for weight loss that they had used successfully when they were younger did not produce the same results at older ages: 'When you're younger, the weight goes much quicker than when you're older. … If I went a week eating the way I ate last week I would have lost half a stone, no problem. I lost a pound. It's much, much more difficult at this age to lose weight, it really is' (106, female, 71 years, BMI 24.2) and 'I eat less than I used to and it doesn't seem to make much difference. … You know, [at younger ages] when you said you would lose weight and you cut down on something, it showed. But I don't think it makes much difference when you grow older' (103, female, 69 years, BMI 26.1).

### Limited guidance on weight management from health professionals

Participants discussed the role of health professionals in their weight management efforts, with mixed experience. Many felt it was important for health professionals to be involved with older people's weight management efforts: 'I think the doctor's is the first stop for somebody if they're very overweight. You have to tell them point blank: 'You're just too heavy and you're going to suffer for it, you won't live very long like this'. You have to be cruel to be kind' (107, female, 78 years, BMI 28.0) and thought older people concerned about their weight should 'Talk to your general practitioner (GP), find out if you need to

lose weight, if so do it properly—either through a GP or a dietitian' (109, female, 72 years, BMI 26.4).

Some participants said their doctor had told them not to worry about being overweight: 'My doctors years ago were saying to me, 'You're a little bit overweight, but it's regular [stable]. That's good; if you're 11 stone and two years later you're still at 11 stone that's a good sign" (111, male, 84 years, BMI 27.0) and 'When I went last time for my regular check up, [the GP] weighed me and said 'Oh, you're so many kilos, that's ok'. I thought 'you must be joking!' If you sort of double that, multiply that by 2.2 or whatever it is [to convert into pounds], that's a hell of a lot of weight! And she didn't seem bothered' (105, male, 72 years, BMI 28.9).

A number of participants had received guidance and support for weight management from a health professional. Most found it useful: 'The doctor asked me, did I want to go for prescription exercise. I agreed to that. … When I was out of breath, he told me it had got a lot to do with my weight—and that I was smoking—but it was to do with weight. Once I went there and I got into the programme and [my weight] started to come down I was fine' (109, female, 72 years, BMI 26.4) and 'I went privately to somebody—we've got a small insurance thing—and, he did these tests and I had an ulcer. And my weight had gone up a lot, it appeared, at the time. And he just politely said, 'It would help if you lost weight.' He was just a very nice man, very gentle. So I lost quite a bit, and I managed to get it down' (112, female, 70 years, BMI 23.3). However, some reported a less positive experience: 'I did go to see an NHS health coach for weight, but I found it really useless. … It was so basic, I got bored with it. Or they give you reams of information to read but it doesn't actually stay' (110, female, 71 years, BMI 40.4).

Some participants mentioned a total lack of advice from health professionals: 'I have to go to the doctor every six months, blood tests and everything. … They don't get on [at] me and I wonder why, because I know I'm really overweight, but they don't seem to make a lot of comment. They just say 'Oh you should cut out the biscuits', they don't say, 'Do this, do that.' It really surprises me' (105, male, 72 years, BMI 28.9) and 'When you see somebody professional they just don't do anything about it. They could be a bit more helpful I think' (112, female, 70 years, BMI 23.3). Others reported simply having been 'told off' about their weight: 'I get told off by the doctor for putting on weight: 'I think it should be a bit less than that,' she will say. She doesn't like me putting on too much weight' (111, male, 84 years, BMI 27.0) and 'I had a stomach ulcer, and you get acidity in the back of your throat. So I did talk it over with the senior GP and she said, 'Oh well, if you're overweight then you lose it or you put up with it.' That was it' (112, female, 70 years, BMI 23.3).

A number of participants questioned how much doctors could actually do to help: 'Old people go straight to the doctor which is a bit sad really, because what can the doctor do? He would probably give you a diet sheet. But as soon as I'm told what to eat, I don't want to eat it'

(107, female, 78 years, BMI 28.0). Others felt that advice from a doctor was (or would be) a powerful motivator: '[The GP] said about going back to the gym and that has spurred me on' (114, female, 70 years, BMI 36.5) and 'You know, if someone at the doctors' had said, 'Do that,' it would have been done' (105, male, 72 years, BMI 28.9). Some participants highlighted the lack of clear information on weight loss at older ages: 'There is a lack of information; I haven't seen anything that is targeting older people. They just say 'Eat less and move more.' I think it's important to have information about what to eat, and to have some supportive groups targeting the over-60s' (110, female, 71 years, BMI 40.4) and 'I don't know who I'd ask. The doctor's not interested. I could see somebody privately but it costs, and are they going to be any better really?' (112, female, 70 years, BMI 23.3).

## DISCUSSION

The results of this qualitative study indicate that weight remains an issue of significant concern to many older people with recent experience of weight management, but is less important to others. Although participants felt it was more socially acceptable to have excess weight at older than younger ages, they reported substantial negative impacts on physical and psychological health associated with carrying excess weight that they felt could be alleviated by weight loss. Participants described how losing weight had become more difficult as they had got older, citing a range of barriers to effective weight management including age-related declines in health and mobility, emotional factors such as comfort eating and lack of will-power, changes in the food environment and the impact of retirement on food intake and physical activity. While the majority indicated that they would appreciate support from a health professional in managing their weight, few reported having received useful advice in the past.

This study has a number of limitations. The sample size was small but comparable with other qualitative studies,[42 43] and we believe data were saturated as no new themes emerged from the last interviews. It is likely that saturation was achieved with a relatively small sample size as a result of the homogeneous nature of the sample. The generally positive attitudes towards weight management may be explained by self-selection bias. Those interested in our study may be those with a long-term interest in weight control, or those who have become interested as they have got older. In order to collect data on experiences as well as attitudes, we restricted our sample to people with recent experience of weight management. Excluding individuals who had not attempted to control their weight in the last 5 years may mean our results overlook the views of those who feel weight management is inappropriate at older ages. Future research should seek to explore this in more depth. Information on weight status was captured using a crude measure of BMI which has been criticised for being an inadequate measure of obesity, particularly among older people who tend to have less muscle mass.[48] We did not ask

participants about the presence of comorbid health problems, which may increase their motivation to lose weight for health reasons, or their social networks, which may influence the extent to which they feel social pressure to lose weight or engage in leisure activities. All participants were white, so our findings do not reflect the views of other ethnic groups in which attitudes towards weight and ageing may differ. We lacked sufficient numbers for subgroup comparisons. Future work could explore the extent to which our findings differ according to age (within the older age range), sex and extent of overweight. In addition, we did not assess socioeconomic status, limiting insight into differences into attitudes towards weight loss across the social gradient. Given that our findings indicated that affordability, leisure time and consumer culture were all important factors, it is likely that people's experiences of managing their weight would vary according to their socioeconomic position. To be inclusive of individuals who were unable to travel to the university, this study used a combination of face-to-face and telephone interviews. However, mixing such methods can also be viewed as a limitation as telephone interviews tend to be shorter than those conducted in person, and typically see interviewees and interviewers speak for less and greater time respectively.[49] This may result in telephone data lacking the same breadth of coverage and depth of detail that can be achieved face-to-face.

Our participants generally reported feeling less social pressure to lose weight than they had when they were younger. Previous research has shown that older women may be less influenced by, and feel less pressure to attain, the media's portrayal of cultural ideals for beauty and thinness,[50 51] and tend to endorse a more curvaceous body shape ideal than their younger counterparts.[52] Some commented that they perceived obesity among older adults to be less stigmatised, consistent with surveys from the UK and USA showing an age-related decline in reports of weight-related discrimination.[53 54] While these observations might lead to the assumption that older adults are less concerned about managing their weight than younger people, this did not appear to be true for our participants. Appearance remained a leading motivator for weight loss for many participants, and an important consideration for others who cited health as their primary concern. This is concordant with the existing literature which indicates that older people still care about their appearance.[37 39] Body dissatisfaction is evident in midlife[33–36] and may even increase as older people gain weight and the discrepancy between their ideal and actual body image widens.[38]

In addition to appearance, health reasons featured highly among our participants' reasons for wanting to lose weight, and for many had become increasingly important as they had got older. This is unsurprising, given that a host of health problems become increasingly prevalent with age.[8] For some participants, fear of developing chronic diseases such as diabetes or cancer or experiencing an acute event like a heart attack or stroke was driving their weight loss efforts. Seeing evidence of the adverse health consequences of carrying excess weight had prompted others to think

more seriously about their weight, with many reporting a desire to alleviate the side effects of existing health problems and reduce the need for medication.

Our participants believed that losing weight could lead to substantial improvements in psychological and physical health. No concerns about potential adverse health consequences of weight loss were raised, although one participant felt that some excess weight could be protective in the event of illness. However, without exception, it was reported that weight management becomes increasingly difficult with age. The same was previously reported in a qualitative study exploring older women's perceptions of ideal body weights.[55] In our study, many participants described how weight loss strategies they had used successfully when they were younger no longer yielded the same results. Health problems and loss of stamina and mobility were commonly cited as barriers to effective weight management, as were a host of factors linked to retirement. For the majority, retiring had reduced the need to be physically active and increased time available for food-focused social occasions and sedentary activities, making it more difficult to achieve energy balance. Previous studies using cohort data have observed similar, showing that retirement is associated with a loss of work-related physical activity that is not fully compensated for by leisure-time physical activity,[56 57] increased time spent watching television,[58 59] and increases in body weight and waist circumference in people retiring from active jobs.[60 61] A reduction in income after retirement was also highlighted as a barrier to weight management. Concerns about affordability are commonly reported by those of low socioeconomic status who want to lose weight.[62] By placing additional financial burden on many older people, retirement may reduce their ability to achieve lifestyles pursuant to healthy ageing and exacerbate health inequalities attributable to overweight and obesity. Loneliness was also mentioned as a barrier to weight loss, with some people using food as a source of comfort. Loneliness increases with age[63] and is a common cause of emotional eating, a highly prevalent behaviour among people affected by obesity.[64]

Health professionals were viewed to have an important role to play in guiding and supporting older people's weight management efforts, but participants' experiences with seeking and receiving professional help were varied. While some had received support for weight management from a GP or other health professional, others reported receiving no advice at all—even when they had sought it out—or a 'telling off' about their weight with no guidance as to how to address the issue. Previous research has found that health professionals experience numerous barriers to providing weight loss advice, including perceived lack of time, training, knowledge and confidence.[65–71] In addition, GPs tend to underestimate BMI and weight status which makes them less likely to intervene and discuss weight.[72] Consistent with evidence that weight loss advice from a health professional in primary care is associated weight loss attempts,[40 73] our participants felt that health professional advice would be a powerful motivator and the majority said they would welcome such advice. However, it is important to note that

these findings only reflect the views of older people who have tried to manage their weight, and may not extend to those less concerned about their weight. It is possible that being given unsolicited advice to lose weight may have a negative impact on overall well-being, with people feeling stigmatised, embarrassed and therefore avoiding accessing health services in future.[74]

Nonetheless, while health professionals may be reluctant to encourage weight loss in older patients due to concerns about associated risks (eg, loss of muscle mass and bone density),[24 25] reviews of the literature indicate that weight loss can have significant benefits for physical function and metabolic and cardiovascular risk[31] and that, on balance, weight loss appears advisable.[32] Our findings indicate that weight management is an issue of considerable relevance and importance to many older people, and one for which they would like to receive advice and support. There is therefore a need for further research to identify ways in which health professionals can best support older patients who want to do so to address weight issues while minimising risk to health.

In conclusion, our findings indicate that while many older adults with recent experience of weight management consider weight to be an important issue which has a significant impact on physical and mental health, others are less concerned about losing weight than they were when they were younger. Weight loss is desired in order to improve appearance and, increasingly importantly, health, but a range of barriers and limited support from health professionals make it harder to lose weight successfully at older ages. Health professionals should be encouraged to broach the issue of weight management with older patients and offer guidance and support to those who want it. The development of materials—for both health professionals and older adults—providing information on losing weight safely at older ages and ways of managing issues relating to retirement and declining health and mobility could help address the unmet needs of older people who want help to manage their weight.

**Twitter** @DrSarahEJackson

**Contributors** SEJ and RJB were responsible for the study concept and design. LH acquired the data. SEJ, LH and RJB analysed and interpreted the data. SEJ drafted the manuscript, and all authors revised it for important intellectual content. All the authors had final approval of the version to be published.

**Funding** This work was supported by a research grant from the Cancer Research UK (ref no. C1418/A14133). RJB is supported by Yorkshire Cancer Research University Academic Fellowship Funding.

**Competing interests** None declared.

**Patient consent for publication** Not required.

**Ethics approval** Ethical approval was granted by the University College London Research Ethics Committee, reference 5234/001.

**Provenance and peer review** Not commissioned; externally peer reviewed.

**Data sharing statement** Anonymised interview transcripts can be obtained by the corresponding author on reasonable request.

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
