## [Reviewer comments · BMJ Open]

ARTICLE DETAILS

TITLE (PROVISIONAL)	'Just because I'm old it doesn't mean I have to be fat': A qualitative study exploring older adults' views and experiences of weight management
AUTHORS	Jackson, Sarah; Holter, Linn; Beeken, Rebecca

VERSION 1 – REVIEW

REVIEWER	Claudia Kimie Suemoto University of Sao Paulo Medical School, Brazil
REVIEW RETURNED	13-Aug-2018

GENERAL COMMENTS	The manuscript is interesting and well-written. I am not familiar with qualitative research, and for me, further details about saturation is needed. Since the low number of participants is a concern, I would like to read further details about the how the authors decided to collect data from only 15 people. It would be interest to know if the results are different according to some characteristics of the sample. For example, between men and women, obese vs. overweight, olde vs. very old.
---

REVIEWER	Ken Fujioka Scripps Clinic, USA
REVIEW RETURNED	17-Aug-2018

GENERAL COMMENTS	Very unusual paper using Themic analysis. A Themic analysis is not commonly used in this type of journal and the average reader will not know what it is. Methods wise it is a weak study as the study population was poorly chosen. Its a geriatric study and you say subjects were to be 65 or older but you had a range down to 60 year old subjects. It would be recommended that if you are to choose pts that you describe them better. Example you choose 15 pts but it is unknown if they have medical problems related to weight such as diabetes (by odds several of them did) and they may be more interested in losing weight for health reasons. Unfortunately we do not know the subjects co-morbid problems. When you read the paper there are so many quotes from the subjects that it becomes a difficult read and does not appear to be very scientific.
--

REVIEWER	Dr Bethany Simmonds University of Portsmouth, United Kingdom.
REVIEW RETURNED	27-Nov-2018

GENERAL COMMENTS	While this paper reports on an important and interesting area; the health of older people and notions of obesity, the limitation of the
---

	sample is not fully iterated in the discussion and subsequent conclusion and recommendations are not justified. 1. The sample population firstly is small for a qualitative study. 20 in-depth interviews would usually be considered acceptable from which to make recommendations about specific group. However, recommendations of this kind can not be made based on 15 interviews (the lengths of which is not known), mainly white women (social class unknown), who have all tried to manage their weight and have self-selected to take part. This group will have very specific views on the types of support they would like and they would more than likely of course think losing weight in later life is important, as they have recently attempted to do so. Throughout the discussion, the specificity of the findings on every point and recommendation in terms of health professional advice must be reiterated. Older people who have not attempted to manage their weight may have the opposite views, i.e. that being approached by a health professional may significantly negatively affected their overall wellbeing, may feel stigmatised, embarrassed, and thus may avoid accessing health services in the future. Evidence by Emma Rich (University of Bath) states that children at school being regulated by BMI via health pedagogic material is counterproductive, leading to worsening overall health and wellbeing. Linked to the above, BMI has been problematised by many social, health and exercise scientists for being an inadequate measure of obesity, especially in older people, who have less muscle mass, let alone its limitations on ability to 'measure fat'. This contextual information about this blunt measure should be evident in this article. 2. Social class is also a factor not mentioned in the paper at all, especially when the findings speak to issues of affordability of weight watchers, leisure time, consumer culture. This is a key structural factor shaping older people's ability to consume 'successful/healthy ageing' lifestyles. 3. The sampling strategy of 'data saturation' after 15 interviews could be based on the homogeneous nature of the sample. Furthermore, it states on p. 7, line 8 that the target number of interviews was 15, which suggests quota sampling, not sampling based on data saturation. 4. Participant involvement in this project is weak, the potential to involve older people from a diverse range of backgrounds in the production of a topic guide was there, perhaps via an initial focus group(s). Furthermore, arguing public approval because local groups advertised the flyers, is also weak. 5. Throughout the findings section, data contradicting the theme under which it was placed was evident. For example, on page 8, lines 32-49 and page 9 lines 3-17, these two paragraphs discuss participants who don't think losing weight in later life is worth it and is under the theme entitled 'weight remains a concern at older ages'. If at least eight out of 15 have opposing views, it would suggest the theme doesn't accurately capture an overwhelming
--	--

	theme, some older people (even those from a sample as limited as above) state that weight does not concern them, perhaps this theme is being forced deductively, rather than emerging inductively. Similarly, under the theme 'Excess weight is more acceptable at older than younger ages' on page 9 lines, 29-30, the participant states 'You do it because you feel you should do it' indicating there is a social pressure to lose weight in later life and it isn't necessarily more acceptable. 6. The following themes are under-analysed and overlap each other: Excess weight has a negative impact on health and wellbeing; appearance and health are important motivators for weight loss; excess weight has a negative impact on health and wellbeing; losing weight gets harder as you get older. a) Excess weight has a negative impact on health and wellbeing Depression and self-loathing of their bodies are discussed, but nothing is said in relation to societal pressure regarding youth culture, ageism, sexism, media images, successful ageing etc. At the bottom of this section there is a paragraph which should be located under the first theme (weight remains a concern for older people) as it discussed the protected (proven) effect that being overweight has on health outcomes (particularly, as indicated in relation to osteoporosis, but also other health conditions). b) Appearance and health are important motivators for weight loss This theme encompasses two entirely different motivations which should be separated out into two themes. Health and appearance are not mutually exclusive and need to be teased out in any analyses. Furthermore, on page 11, lines 0-43 the quote about osteoporosis diagnosis is very unclear, particularly in light of section 6 a) above. c) Losing weight gets harder as you get older This 'theme' has subthemes of 'health-related barriers, emotional barriers and situational barriers' which are not discrete from the preceding themes. There is a lot of overlap in points being made. This is confusing as a reader and needs to be re-analysed to consolidate commonalities across narratives and re-developing coherent and discrete themes. This is perhaps due to the absence of 'motivators' often become 'barriers'. Nevertheless, under analysis of 'emotional barriers' and 'situational barriers', where is the mention of sequestration of older people, due to forced retirement, ageism in the workplace, austerity and a lack of health and social care, let alone the impact this has on leisure and social networks. Social structure is almost completely ignored in this paper and the impact this has on the experiences of older people living in the UK. Furthermore, the impact of late modernity on food production, advertising, nutrition, activity levels (McDonaldisation). As a result of the points made the following sentences in the discussion/conclusion need to be amended: p17 lines 15-20; p19. lines 46-54 and page 20 lines 3-5.
--	---

VERSION 1 – AUTHOR RESPONSE

Reviewer: 1

Reviewer Name: Claudia Kimie Suemoto

Institution and Country: University of Sao Paulo Medical School, Brazil Please state any competing interests or state 'None declared': None declared

Please leave your comments for the authors below The manuscript is interesting and well-written. I am not familiar with qualitative research, and for me, further details about saturation is needed. Since the low number of participants is a concern, I would like to read further details about the how the authors decided to collect data from only 15 people.

Response: We have clarified what is meant by saturation:

“We aimed to recruit until it was felt that saturation had been reached (i.e. sampling more data would not provide more information or insights related to our research questions).”

It was this process of determining saturation that led us to the decision to stop recruiting after 15 interviews.

It would be interest to know if the results are different according to some characteristics of the sample. For example, between men and women, obese vs. overweight, olde vs. very old.

Response: Given the small sample size we lack sufficient participant numbers to facilitate comparisons between subgroups. We have added this as a limitation:

“We lacked sufficient numbers for subgroup comparisons. Future work could explore the extent to which our findings differ according to age (within the older age range), sex, and extent of overweight.”

Reviewer: 2

Reviewer Name: Ken Fujioka

Institution and Country: Scripps Clinic, USA Please state any competing interests or state 'None declared': None

Please leave your comments for the authors below Very unusual paper using Themic analysis. A Themic analysis is not commonly used in this type of journal and the average reader will not know what it is.

Response: We acknowledge that many of the journal's readers may not be familiar with thematic analysis, so have tried to provide a coherent description of the method along with relevant references:

“Data were analysed using thematic analysis, a qualitative method for identifying, analysing and reporting themes [45]. This method was chosen with the aim of providing a rich description of the data, and to identify themes at an explicit level using a realist approach [45]. As data were acquired, SEJ read the transcripts for essential familiarisation and to generate initial codes. These were amended and refined through discussion between the researchers in an iterative process until a final coding list was agreed. SEJ coded all the transcripts according to the established coding structure in NVivo V.10 and a random selection of transcripts (n=2) were coded by a second researcher (RJB) to check for reliability. Inter-rater reliability for the coding was generally high (81-100% agreement; mean 98%) with any minor differences resolved by discussion. SEJ and RJB collated codes into common themes, which were reviewed and refined, named and each given a written description. All themes were checked against the transcripts to ensure that they reflected the majority of participants. Individual experiences were also highlighted.”

Methods wise it is a weak study as the study population was poorly chosen. Its a geriatric study and you say subjects were to be 65 or older but you had a range down to 60 year old subjects.

Response: Thank you for highlighting the discrepancy in the age range of our sample. We checked the data and noticed that one participant's age had been incorrectly entered as 60 rather than 69. We

have corrected this and updated the range, mean and standard deviation for age in the text and tables.

It would be recommended that if you are to choose pts that you describe them better. Example you choose 15 pts but it is unknown if they have medical problems related to weight such as diabetes (by odds several of them did) and they may be more interested in losing weight for health reasons. Unfortunately we do not know the subjects co-morbid problems.

Response: Unfortunately, this information was not collected from participants. We have added this limitation to our discussion:

“We did not ask participants about the presence of comorbid health problems, which may increase their motivation to lose weight for health reasons.”

When you read the paper there are so many quotes from the subjects that it becomes a difficult read and does not appear to be very scientific.

Response: This is typical of qualitative papers using the same methodology. In such studies, the transcripts, from which the quotes are taken, are the data. We would therefore like to retain these examples to highlight how our themes were informed by our data.

Reviewer: 3

Reviewer Name: Dr Bethany Simmonds

Institution and Country: University of Portsmouth, United Kingdom.

Please state any competing interests or state 'None declared': None declared.

Please leave your comments for the authors below While this paper reports on an important and interesting area; the health of older people and notions of obesity, the limitation of the sample is not fully iterated in the discussion and subsequent conclusion and recommendations are not justified.

1. The sample population firstly is small for a qualitative study. 20 in-depth interviews would usually be considered acceptable from which to make recommendations about specific group. However, recommendations of this kind can not be made based on 15 interviews (the lengths of which is not known), mainly white women (social class unknown), who have all tried to manage their weight and have self-selected to take part.

Response: We appreciate the sample size is somewhat smaller than might usually be expected for a qualitative study. However, we planned a priori to recruit until we reached saturation and we believed that was the case after 15 interviews. As this reviewer highlights in comment 3, it is possible that we reached saturation in a smaller than average number of interviews due to the homogenous nature of the target population. We now comment on this in the discussion:

“It is likely that saturation was achieved with a relatively small sample size as a result of the homogeneous nature of the sample.”

Our ability to recruit more widely was limited by the resources available for this study, equally qualitative research is not set up to generalise to the population. However, we raise the lack of ethnic diversity and self-selection as limitations in our discussion:

“All participants were white, so our findings do not reflect the views of other ethnic groups in which attitudes towards weight and ageing may differ.”

“The generally positive attitudes towards weight management may be explained by self-selection bias. Those interested in our study may be those with a long-term interest in weight control, or those who have become interested as they have got older.”

We have also added a sentence on the lack of insight into the socioeconomic status of our participants:

“In addition, we did not assess socioeconomic status, limiting insight into differences into attitudes towards weight loss across the social gradient.”

This group will have very specific views on the types of support they would like and they would more than likely of course think losing weight in later life is important, as they have recently attempted to do so. Throughout the discussion, the specificity of the findings on every point and recommendation in terms of health professional advice must be reiterated.

Response: We have edited the discussion to ensure it is clear that these findings refer to older people with recent experience of weight management.

Older people who have not attempted to manage their weight may have the opposite views, i.e. that being approached by a health professional may significantly negatively affected their overall wellbeing, may feel stigmatised, embarrassed, and thus may avoid accessing health services in the future. Evidence by Emma Rich (University of Bath) states that children at school being regulated by BMI via health pedagogic material is counterproductive, leading to worsening overall health and wellbeing.

Response: We thank the reviewer for raising this issue. We have added to our discussion as follows: “However, it is important to note that these findings only reflect the views of older people who have tried to manage their weight, and may not extend to those less concerned about their weight. It is possible that being given unsolicited advice to lose weight may have a negative impact on overall wellbeing, with people feeling stigmatised, embarrassed, and therefore avoiding accessing health services in future [74].”

Linked to the above, BMI has been problematised by many social, health and exercise scientists for being an inadequate measure of obesity, especially in older people, who have less muscle mass, let alone its limitations on ability to 'measure fat'. This contextual information about this blunt measure should be evident in this article.

Response: We now acknowledge this as a limitation: Information on weight status was captured using a crude measure of BMI, which has been criticised for being an inadequate measure of obesity, particularly among older people who tend to have less muscle mass [48].

2. Social class is also a factor not mentioned in the paper at all, especially when the findings speak to issues of affordability of weight watchers, leisure time, consumer culture. This is a key structural factor shaping older people's ability to consume 'successful/healthy ageing' lifestyles.

Response: We have added to the discussion in a couple of places highlighting this issue:

“In addition, we did not assess socioeconomic status, limiting insight into differences into attitudes towards weight loss across the social gradient. Given that our findings indicated that affordability, leisure time and consumer culture were all important factors, it is likely that people's experiences of managing their weight would vary according to their socioeconomic position.”

“A reduction in income after retirement was also highlighted as a barrier to weight management. Concerns about affordability are commonly reported by those of low socioeconomic status who want to lose weight [62]. By placing additional financial burden on many older people, retirement may reduce their ability to achieve lifestyles pursuant to healthy ageing and exacerbate health inequalities attributable to overweight and obesity.”

3. The sampling strategy of 'data saturation' after 15 interviews could be based on the homogeneous nature of the sample. Furthermore, it states on p. 7, line 8 that the target number of interviews was 15, which suggests quota sampling, not sampling based on data saturation.

Response: We have added the point about saturation being reached with a relatively small sample in our discussion:

“It is likely that saturation was achieved with a relatively small sample size as a result of the homogeneous nature of the sample.”

We have edited the statement about the anticipated sample size to clarify that this was not a target number:

“Based on previous qualitative studies in similar groups, we expected that around 15 participants would be required to achieve this [42,43], although no target number of participants was set.”

4. Participant involvement in this project is weak, the potential to involve older people from a diverse range of backgrounds in the production of a topic guide was there, perhaps via an initial focus group(s). Furthermore, arguing public approval because local groups advertised the flyers, is also weak.

Response: We acknowledge more could have been done to involve older people in the design of the study. It was originally a student project and our resources were limited. We have removed the sentence on public approval.

5. Throughout the findings section, data contradicting the theme under which it was placed was evident. For example, on page 8, lines 32-49 and page 9 lines 3-17, these two paragraphs discuss participants who don't think losing weight in later life is worth it and is under the theme entitled 'weight remains a concern at older ages'. If at least eight out of 15 have opposing views, it would suggest the theme doesn't accurately capture an overwhelming theme, some older people (even those from a sample as limited as above) state that weight does not concern them, perhaps this theme is being forced deductively, rather than emerging inductively.

Response: We thank the reviewer for highlighting this oversight. We have renamed this theme 'mixed views on the importance of weight at older ages' to more accurately capture what is reflected in the data.

Similarly, under the theme 'Excess weight is more acceptable at older than younger ages' on page 9 lines, 29-30, the participant states 'You do it because you feel you should do it' indicating there is a social pressure to lose weight in later life and it isn't necessarily more acceptable.

Response: The full quote reads “I think if you're older you [lose weight] for yourself, you don't do it to satisfy people around you, to be liked more. You just do it because you want to lose weight. ... You do it because you feel you should do it, and for your own benefit, that's all”. We do not believe this implies that there is social pressure to lose weight, rather it is the person's own desire to lose weight that is driving it.

6. The following themes are under-analysed and overlap each other: Excess weight has a negative impact on health and wellbeing; appearance and health are important motivators for weight loss; excess weight has a negative impact on health and wellbeing; losing weight gets harder as you get older.

a) Excess weight has a negative impact on health and wellbeing

Depression and self-loathing of their bodies are discussed, but nothing is said in relation to societal pressure regarding youth culture, ageism, sexism, media images, successful ageing etc.

Response: The results we describe were themes that emerged from analysis of the interviews. While we appreciate that youth culture, ageism, sexism, media images, and successful ageing would indeed be relevant components of this theme, they were not mentioned by participants so it would be inappropriate to discuss them here.

At the bottom of this section there is a paragraph which should be located under the first theme (weight remains a concern for older people) as it discussed the protected (proven) effect that being

overweight has on health outcomes (particularly, as indicated in relation to osteoporosis, but also other health conditions).

Response: We have moved this paragraph to the first theme, as suggested.

b) Appearance and health are important motivators for weight loss

This theme encompasses two entirely different motivations which should be separated out into two themes. Health and appearance are not mutually exclusive and need to be teased out in any analyses.

Response: We have separated discussion of appearance and health into two separate themes.

Furthermore, on page 11, lines 0-43 the quote about osteoporosis diagnosis is very unclear, particularly in light of section 6 a) above.

Response: We have deleted this quote.

c) Losing weight gets harder as you get older

This 'theme' has subthemes of 'health-related barriers, emotional barriers and situational barriers' which are not discrete from the preceding themes. There is a lot of overlap in points being made. This is confusing as a reader and needs to be re-analysed to consolidate commonalities across narratives and re-developing coherent and discrete themes. This is perhaps due to the absence of 'motivators' often become 'barriers'. Nevertheless, under analysis of 'emotional barriers' and 'situational barriers', where is the mention of sequestration of older people, due to forced retirement, ageism in the workplace, austerity and a lack of health and social care, let alone the impact this has on leisure and social networks. Social structure is almost completely ignored in this paper and the impact this has on the experiences of older people living in the UK.

Response: We have edited the wording of our discussion of the health-related barriers to weight loss to highlight the contrast between this and the finding that health concerns were a strong motivator (as outlined in an earlier theme):

“While health was a commonly reported motive for weight loss, specific problems with health and mobility were also mentioned as barriers to weight loss at older ages:...”

Participants did not talk about forced retirement, ageism in the workplace, austerity or a lack of health and social care, so while they are certainly relevant issues these are not discussed under this theme.

We now comment on the lack of information about social networks in our discussion of the study's limitations:

“We did not ask participants about ... their social networks, which may influence the extent to which they feel social pressure to lose weight or engage in leisure activities.”

Furthermore, the impact of late modernity on food production, advertising, nutrition, activity levels (McDonaldisation).

Response: While participants did not bring up changes in food production or the need for physical activity brought about by modern innovations, we discuss their comments on food availability and nutrition knowledge:

“While most explanations for weight loss becoming more difficult over time related to the ageing process, some participants attributed increasing difficulty in managing their weight to changes in the food environment: “I think one of the big changes is [that now] you are bombarded with food. When I was growing up, you'd never think of going to get a takeaway. Where I live, on the high street, at every corner is something, it's all fast food” (110, female, 71 years, BMI 40.4) and “I lived in a house where there wasn't actually a huge amount of food at some points when I was small, so not having enough to eat was relatively common” (101, female, 67 years, BMI 28.2). However, some commented

that information on healthy living was more readily available than when they were younger: “There was no information at all on what made you fat; nobody said anything about it (110, female, 71 years, BMI 40.4) and “Everybody, absolutely everybody knows [what they should be eating]; I can’t believe that anybody on the planet who doesn’t know. The five a day message has certainly got around” (113, female, 71 years, BMI 35.7), and one participant mentioned that better availability of and access to healthy foods had made it easier to lose weight: “I think it’s better now, because I’m eating better things. And when I was trying to lose weight before there wasn’t the variety of food available” (112, female, 70 years, BMI 23.3).”

As a result of the points made the following sentences in the discussion/conclusion need to be amended:

p17 lines 15-20; p19. lines 46-54 and page 20 lines 3-5.

Response: We have edited these accordingly:

“The results of this qualitative study indicate that weight remains an issue of significant concern to many older people with recent experience of weight management, but is less important to others. Although participants felt it was more socially acceptable to have excess weight at older than younger ages, they reported substantial negative impacts on physical and psychological health associated with carrying excess weight that they felt could be alleviated by weight loss. Participants described how losing weight had become more difficult as they had got older, citing a range of barriers to effective weight management including age-related declines in health and mobility, emotional factors such as comfort eating and lack of willpower, changes in the food environment, and the impact of retirement on food intake and physical activity. While the majority indicated that they would appreciate support from a health professional in managing their weight, few reported having received useful advice in the past.”

“In conclusion, our findings indicate that while many older adults with recent experience of weight management consider weight to be an important issue which has a significant impact on physical and mental health, others are less concerned about losing weight than they were when they were younger. Weight loss is desired in order to improve appearance and, increasingly importantly, health, but a range of barriers and limited support from health professionals make it harder to lose weight successfully at older ages. Health professionals should be encouraged to broach the issue of weight management with older patients and offer guidance and support to those who want it. The development of materials – for both health professionals and older adults – providing information on losing weight safely at older ages and ways of managing issues relating to retirement and declining health and mobility could help address the unmet needs of older people who want help to manage their weight.”

VERSION 2 – REVIEW

REVIEWER	Claudia Kimie Suemoto University of Sao Paulo, Brazil
REVIEW RETURNED	20-Dec-2018

GENERAL COMMENTS	The manuscript improved significantly after review. I do not have further questions.
---

REVIEWER	Dr Bethany Simmonds The University of Portsmouth, England.
REVIEW RETURNED	19-Dec-2018

GENERAL COMMENTS	The authors have addressed my concerns adequately. I am satisfied with the amendments. Thank you.
---